# Macedon: Minimizing Representation Coding Rate Reduction for Cross-Lingual Natural Language Understanding

**Haoyu Wang[§], Yaqing Wang[†], Huaxiu Yao[*], and Jing Gao[§]**

[§]Purdue University, West Lafayette, IN, USA
[†]Google Research, New York, NY, USA
[*]UNC-Chapel Hill, Chapel Hill, NC, USA
[§]{wang5346,jinggao}@purdue.edu,
[†]yaqingwang@google.com,
[*]huaxiu@cs.unc.edu

## Abstract

Cross-lingual natural language understanding (NLU) is one of the fundamental tasks of NLP. The goal is to learn a model which can generalize well on both high-resource and low-resource language data. Recent pre-trained multilingual language models, e.g., multilingual BERT, XLM, have shown impressive performance on cross-lingual NLU tasks. However, such promising results request the use of sufficient training data, which is a difficult condition to satisfy for low-resource language.When the data is limited in those low resource languages, the accuracy of existing models will drop. In light of this challenge, we investigate the important task of how to train the cross-lingual model with abundant high-source language data and limited low-resource language data. Existing methods typically learn language-agnostic representation via adversarial training and mutual information estimation. Existing approaches may suffer When data is very limited (e.g., low-resource language) because it is challenging to estimate data distribution accurately. To tackle this issue, we propose a conceptually innovative approach to remove language-associated information via **m**inimizing represent**a**tion **c**oding rate r**ed**ucti**on** (Macedon). Specifically, Macedon avoids using extra codes to encode language-related information, which is measured by the rate-distortion function. To validate the effectiveness of Macedon, we conduct extensive experiments on three tasks, including paraphrase identification, natural language inference, and query advertisement matching. The experiment results show that the proposed Macedon outperforms state-of-the-art cross-lingual NLU approaches.

## 1 Introduction

Globally, there are about 7,106 living languages spoken[1], but only 91 languages have at least 10 million first language speakers counted in 2022[2]. Therefore, most languages (i.e., low-resource languages) have limited data when considered in natural language understanding (NLU) tasks (Joshi et al., 2020), which covers a variety of sub-tasks dealing with machine reading comprehension such as sentiment classification, named entity recognition, and part-of-speech tagging. As a result, models trained on such limited data may have poor performance. An immediate solution one can think of is to collect more labeled data for low-resource languages. However, it could be infeasible due to the prohibitively intensive human annotation process that is both time-consuming and costly. Thus, cross-lingual NLU has gained increasing interests in recent years (Nooralahzadeh et al., 2020; Wang et al., 2021b; Mao et al., 2021; Wu and Dredze, 2019), in which the model is trained with abundant high-resource language data and limited annotated low-resource language data to make the model generalize well for all languages.

Recently, pre-trained multilingual language models (MLMs), such as multilingual BERT (Devlin et al., 2018) and XLM (Lample and Conneau, 2019), have shown promising results in transfer learning across different languages in multiple downstream tasks (Hu et al., 2020; Liang et al., 2020). In MLMs, different languages are able to share one vocabulary table, enabling the model to learn information that is shared among those languages. Typical MLMs, however, require parallel corpora in order to fine-tune downstream tasks, which can be difficult to achieve when there exist low-resource languages. Additionally, it may be difficult to align different languages and yield satisfying transfer performance when there are huge differences between low-resource and high-resource languages in morphology, syntax, or semantics (Ahmad et al., 2019; Hu et al., 2020).

---

[1]https://www.ethnologue.com/

[2]https://en.wikipedia.org/wiki/List_of_languages_by_number_of_native_speakers

Prior studies have shown that aligning different language representations is effective for cross-lingual transfer. The key idea is to enforce similar representations for sentences that share similar semantics across languages. Existing methods fall into two categories. The first line of methods has to be based on parallel corpora. The method aligns different languages in the latent space in accordance with parallel sentences (Yang et al., 2021; Gritta and Iacobacci, 2021; Cao et al., 2020; Pan et al., 2020; Dou and Neubig, 2021). For languages with limited resources, however, parallel corpora are extremely difficult to obtain. Therefore, another line of approaches leverages adversarial training and employs Generative Adversarial Networks (GAN) (Goodfellow et al., 2014) to learn language-independent representations (Chen et al., 2021, 2018b,a; Keung et al., 2019; Lee and Lee, 2019; Wang et al., 2021a; Zou et al., 2018). It is worthwhile to note that adversarial training-based methods rely on an accurate estimation of data distributions, which may be infeasible with limited labeled examples in low-resource languages.

In this paper, we solve the cross-lingual NLU tasks from another perspective. The goal is still to align different languages. However, different from adversarial training, we do not need to align the distributions of different language representations strictly. We propose to relax the condition and simply force different language representations to overlap in the representation space. To achieve this goal, we propose Macedon which **m**inimizes represent**a**tion **c**oding rate r**edu**cti**on**. Macedon is built upon the coding rate function, which measures the minimal average number of bits to encode a set of vectors (Ma et al., 2007; Yu et al., 2020). Specifically, we measure two kinds of coding rates: the coding rate of all language representations and the average coding rate of each language representation. We encourage the two coding rates to be almost the same value. Intuitively, the learned representation does not use extra codes to encrypt language information, and can be considered as language irrelevant approximately. In addition, the coding rate function uses empirical covariance matrix to compute rate-distortion function directly, which is likely to be singular and highly biased. In Macedon, we employ a simple but effective method to modify the empirical covariance matrix to make it positive definite and reduce its bias.

The contributions of the paper are summarized as follows:

- We propose a novel method for cross-lingual NLU tasks. To the best of our knowledge, this is the first attempt to leverage coding rate function in cross-lingual NLU tasks.

- We provide a new perspective of modeling cross-lingual representations. We leverage data compression to mix different language representations in the latent space. It is a more effective way to align different languages compared with adversarial training under limited data.

- We conduct extensive experiments on three public datasets. Results show that the proposed Macedon outperforms the state-of-the-art baselines. Furthermore, experimental results show that the proposed Macedon is able to reduce the distance between different language representation distributions.

## 2 Related Work

### 2.1 Cross-lingual NLU

Cross-lingual NLU tasks has been studied from two major perspectives: 1) multilingual word embedding (Doval et al., 2019; Ormazabal et al., 2019; Ruder et al., 2019; Shi et al., 2015; Søgaard et al., 2019; Wada and Iwata, 2018; Xu et al., 2018; Cao et al., 2020; Dou and Neubig, 2021), and 2) pre-trained multilingual language models, such as multilingual BERT (Devlin et al., 2018), and XLM-RoBERTa (Lample and Conneau, 2019). Some studies (Devlin et al., 2018; Wu and Dredze, 2019) have shown pre-trained multilingual language models (MLM) have much better performance than traditional word embedding methods for multiple NLU tasks. Therefore, recent works (Hu et al., 2020; Liang et al., 2020) usually fine-tune MLMs for downstream NLU tasks. Specifically, works based on MLMs can be roughly categorised as data augmentation-based and language alignment-based methods.

For the data augmentation-based methods, they are to solve the key challenge of cross-lingual NLU: lack of low-resource language data. For example, Code-switch (Qin et al., 2020) proposes to randomly replace the phrase of high-resource languages to corresponding low-resource language phrase; Dong et al. (2021) augments the data via reordering; Ahmad et al. (2021) proposes the augmentation method based on syntax; and Bari et al.

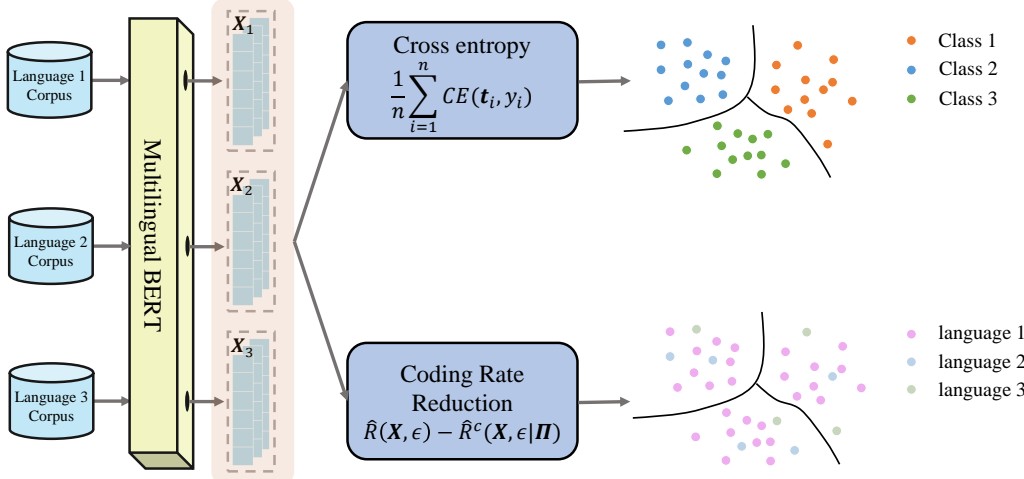

Figure 1: The framework of proposed Macedon.

(2020) generates augmented data directly from MLMs based on the vicinity distribution of high-resource and low-resource language data samples. They are effective to improve cross-lingual NLU performance. However, they usually need to draw on additional resources such as bilingual dictionary.

Another category is to align different languages. Some works attempt to align the contextual word embedding of different languages, such as learning projection transformations (Aldarmaki and Diab, 2019; Wu et al., 2019) and forcing model into having similar predictions for parallel sentences (Yang et al., 2021; Gritta and Iacobacci, 2021; Cao et al., 2020; Pan et al., 2020; Dou and Neubig, 2021). And there are also a lot of works exploring adversarial training to align different languages (Chen et al., 2021, 2018b,a; Keung et al., 2019; Lee and Lee, 2019; Wang et al., 2021a; Zou et al., 2018). Their motivation is to learn language-agnostic representation to make model focus more on understand text semantics and generalize better on low-resource languages. However, this line of work usually needs parallel corpus or large amount low-resource language data, which are expensive to obtain and may be not always available.

### 2.2 Coding Rate-distortion Function for Representation Learning

Coding rate-distortion (Berger, 2003; Ma et al., 2007) is a concept about data compression from information theory. Recently, it is introduced into representation learning. Chan et al. (2021) is the first work using coding rate-distortion function as an objective function for designing interpretable

deep neural networks. Recently, Yu et al. (2020) shows maximizing coding rate reduction can be considered as an alternative to traditional objective function such as cross entropy. Li et al. (2022) and Chowdhury and Chaturvedi (2022) also apply the maximizing coding rate reduction method to manifold clustering and fairness representation respectively. Different from previous work, our proposed Macedon is to minimize coding rate reduction function to learn language-agnostic representation.

## 3 Preliminaries

### 3.1 Rate Distortion for Finite Data

Rate distortion is a concept for lossy compression used in information theory. Given learned representation $\boldsymbol{X} = [\boldsymbol{x}_1, ..., \boldsymbol{x}_m] \in \mathbb{R}^{d \times m}$ and a precision $\epsilon^2$, where $d$ is the representation dimension and $m$ is number of data samples, the rate distortion $R(\boldsymbol{X}, \epsilon)$ is to measure the minimal number of binary bits to encode $\boldsymbol{X}$ averagely with the precision constraint $\mathbb{E}[\|\boldsymbol{x} - \hat{\boldsymbol{x}}\|_2] \leq \epsilon^2$. Yu et al. (2020) and Ma et al. (2007) show that if $\boldsymbol{x}_i$ is sampled from a subspace-like distribution, the precise estimation (tight bound) of $R(\boldsymbol{X}, \epsilon)$ is

$$R(\boldsymbol{X}, \epsilon) = \frac{1}{2} \log \det(\boldsymbol{I} + \frac{d}{m\epsilon^2} \boldsymbol{X} \boldsymbol{X}^T), \quad (1)$$

where $\frac{1}{m} \boldsymbol{X} \boldsymbol{X}^T$ is the empirical covariance matrix of $\boldsymbol{X}$. Intuitively, the $R(\boldsymbol{X}, \epsilon)$ is proportional to the logarithm of the volume of region spanned by $\boldsymbol{X}$. If $\boldsymbol{X}$ is very concentrated, then the volume of region they span will be small, i.e. the value of $R(\boldsymbol{X}, \epsilon)$ will be small, and vice versa.

### 3.2 Rate Distortion for Data with Mixed Distribution

Generally, the representation $\boldsymbol{X}$ may lie in multiple subspaces, e.g. multi-class, which means the representation corresponds to a mixed distribution, i.e. $\boldsymbol{X} = \boldsymbol{X}_1 \bigcup ... \bigcup \boldsymbol{X}_k$, where $\boldsymbol{X}_i$ corresponds to the representation in the class $i$. We use a set of diagonal matrices $\boldsymbol{\Pi} = \{\boldsymbol{\Pi}_j \in \mathbb{R}^{m \times m}\}_{j=1}^k$ to indicate the probability that sample $i$ belongs to $\boldsymbol{X}_j$ by its diagonal entry $\boldsymbol{\Pi}_j[i, i]$. The $\boldsymbol{\Pi}_j$ satisfies that $\sum_i \boldsymbol{\Pi}_j[i, i] = 1$ and $\sum_j \boldsymbol{\Pi}_j = \boldsymbol{I}$. Yu et al. (2020) and Ma et al. (2007) indicate that the rate distortion function for data with such mixed distribution is:

$$R^c(\boldsymbol{X}, \epsilon | \boldsymbol{\Pi})$$
$$= \sum_{j=1}^k \frac{tr(\boldsymbol{\Pi}_j)}{2m} \log \det(\boldsymbol{I} + \frac{d}{tr(\boldsymbol{\Pi}_j)\epsilon^2} \boldsymbol{X} \boldsymbol{\Pi}_j \boldsymbol{X}^T).$$

Here, we just consider that each example belongs to one category.

## 4 Minimizing Representation Coding Rate Reduction

### 4.1 Problem Formulation

The cross-lingual natural language understanding (NLU) aims to learn a model $f(\boldsymbol{t}_i) \rightarrow y_i$ on the training set $\mathcal{D} = \{(\boldsymbol{t}_i, y_i)\}_{i=1}^n$, which consists of large amount of high-resource language (e.g., English) data and limited low-resource language data. Here $\boldsymbol{t}_i$ is the text of training example $i$, $y_i$ is the corresponding label, and $n$ is the number of training examples. In Macedon, we aim to learn a model $f_\theta(\cdot)$ that can generalize well to both high-resource and low-resource languages.

In the next sections, we use $\boldsymbol{X}$ to denote the representation of the whole training data, and use $\boldsymbol{X}_i$ to denote the representation of the $i$-th language in the training set. We also denote $n_i$ as the number of training data for the $i$-th language. In the following, we first overview the entire framework of Macedon and introduce the detail of rate reduction function and the whole loss function in Section 4.3.

### 4.2 Overview

The goal of Macedon is to learn an effective cross-lingual model which can generalize well to all languages, including high-resource languages and limited low-resource languages. The framework of Macedon is shown in Figure 1. It is to align different languages via encouraging them to overlap in the representation space, which can be considered as language irrelevant approximately. The objective function of the proposed Macedon includes two terms: 1) cross entropy loss to classify data accurately, and 2) coding rate reduction loss, which is to make different language representation mixed in the latent space. We introduce the proposed Macedon in the following section in detail.

### 4.3 Coding Rate-Distortion Maximization

To enable the learned representation to be transferable among different languages, we hope the model can focus on understanding the language-agnostic semantics of text following (Chen et al., 2021, 2018b; Keung et al., 2019; Wang et al., 2021a; Zou et al., 2018). In the other word, the sentence representation should preserve semantic information while removing language-related information. Therefore, we propose the following two criteria for the representation learning:

- The learned representation should contain enough information for the NLU task. It should be informative for the target label $y_i$.

- The distribution of representations from different languages should be similar. It is to make it more transferable among different languages.

For the first criterion, it corresponds to a standard classification problem and we minimize the cross entropy between the prediction and label. To satisfy the second criterion, we need to make the representation of different languages mixed and overlap in the representation space. Usually, adversarial training such as Generative Adversarial Network (GAN) can be used to learn feature-invariant representation, which has shown their effectiveness on domain adaptation (Tzeng et al., 2017). However, when there are very limited target domain data, it is difficult to estimate target domain distribution. Therefore, adversarial training and mutual information estimation may be not effective in the cross-lingual setting, which is shown in the Section 5.3. In our paper, we propose an alternative method to solve this problem based on rate distortion. The key motivation is that the average bit number used to encode all language representation almost equals to the average bit number used to encode each language representation respectively. In other words, it nearly does not need to use extra bits to encode (remember) language. To achieve this goal, we control the absolute value of the rate

reduction $R(\boldsymbol{X}, \epsilon) - R^c(\boldsymbol{X}, \epsilon | \Pi)$ to be smaller than a small value $\delta$. Therefore, we loss function can be represented as

$$\min \frac{1}{n} \sum_{i=1}^{n} \text{CE}(\boldsymbol{t}_i, y_i)$$
$$s.t. |R(\boldsymbol{X}, \epsilon) - R^c(\boldsymbol{X}, \epsilon | \Pi)| \leq \delta, \qquad (2)$$

where CE represents the cross entropy loss. Because $R(\boldsymbol{X}, \epsilon) - R^c(\boldsymbol{X}, \epsilon | \Pi) \geq 0$ (Yu et al., 2020), we can rewrite the loss function as

$$\min \frac{1}{n} \sum_{i=1}^{n} \text{CE}(\boldsymbol{t}_i, y_i)$$
$$s.t. R(\boldsymbol{X}, \epsilon) - R^c(\boldsymbol{X}, \epsilon | \Pi) \leq \delta. \qquad (3)$$

However, in cross-lingual setting, the empirical covariance matrix in $R(\boldsymbol{X}, \epsilon)$ and $R^c(\boldsymbol{X}, \epsilon | \Pi)$ may result in large bias because there are very limited samples. Because the representation of multilingual BERT is larger than number of low-resource language training data, their empirical covariance matrix may be even not positive definite. To reduce the bias under such limited data condition, we modify the empirical covariance matrix $\boldsymbol{X}\boldsymbol{X}^T$ as

$$(1 - \alpha)\boldsymbol{X}\boldsymbol{X}^T + \alpha \frac{tr(\boldsymbol{X}\boldsymbol{X}^T)}{d}\boldsymbol{I}, \qquad (4)$$

and it has two great properties: 1) it is a positive definite matrix shown in Proposition 1, and 2) $\hat{R}(\boldsymbol{X}, \epsilon) - \hat{R}^c(\boldsymbol{X}, \epsilon | \Pi) \geq 0$ shown in Proposition 2, where $\hat{R}(\boldsymbol{X}, \epsilon)$ and $\hat{R}^c(\boldsymbol{X}, \epsilon | \Pi)$ are modified rate-distortion functions respectively.

**Proposition 1.** *The matrix* $(1 - \alpha)\boldsymbol{X}\boldsymbol{X}^T + \alpha \frac{tr(\boldsymbol{X}\boldsymbol{X}^T)}{d}\boldsymbol{I}$ *is positive definite.*

The proof is shown in the appendix.

**Proposition 2.** $\hat{R}(\boldsymbol{X}, \epsilon) - \hat{R}^c(\boldsymbol{X}, \epsilon | \Pi) \geq 0$. *The equality holds when* $(1 - \alpha)\frac{\boldsymbol{X}_1\boldsymbol{X}_1^T}{n_1} + \alpha \frac{\boldsymbol{X}_1\boldsymbol{X}_1^T}{d}\boldsymbol{I} = ... = (1 - \alpha)\frac{\boldsymbol{X}_k\boldsymbol{X}_k^T}{n_k} + \alpha \frac{\boldsymbol{X}_k\boldsymbol{X}_k^T}{d}\boldsymbol{I}.$

The proof is shown in the appendix.

Therefore, we can rewrite the objective function as an unconstrained optimization problem:

$$\frac{1}{n} \sum_{i=1}^{n} \text{CE}(\boldsymbol{t}_i, y_i) + \eta(\hat{R}(\boldsymbol{X}, \epsilon) - \hat{R}^c(\boldsymbol{X}, \epsilon | \Pi)),$$

where $\eta$ is a positive hyper-parameter.

## 5   Experiment

In this section, we evaluate the proposed Macedon with the goal of answering the following questions:

RQ1 How does Macedon perform compared to state-of-the-art baselines?

RQ2 What is the relationship between the proposed Macedon and adversarial training?

RQ3 Is the proposed Macedon effective to align different language representations?

RQ4 How does the performance change with respect to different coefficients of the proposed Macedon?

### 5.1   Datasets and Experiment Settings

#### 5.1.1   Datasets

To evaluate the performance of the proposed Macedon comprehensively, we conduct experiments on three public cross-lingual benchmark datasets, including PAWS-X (Hu et al., 2020), QADSM (Liang et al., 2020), and XNLI (Hu et al., 2020). The three datasets correspond to paraphrase identification, query advertisement matching, and natural language inference. For the three datasets, we randomly sample 100 few-shot data for each language other than English. In XNLI, we add an extra experiment using 2.5k data instances as few-shot data for low-resource languages, namely Macedon 2.5k, to compare with the results in (Nooralahzadeh et al., 2020) fairly. We tune hyper-parameters and evaluate the developed models on the pre-split validation and testing set. The statistics of datasets are summarized in Table 2. PAWS-X is public release but the Licence information is not presented in the source website[3]. QADSM and XNLI are available under Creative Commons Attribution 4.0 International and Attribution-NonCommercial 4.0 International Licenses respectively. We follow the licence of datasets for research use.

#### 5.1.2   Baselines

We adopt the following state-of-the-art methods as baselines:

- mbert (Devlin et al., 2018) is the multi-lingual version of BERT (Devlin et al., 2018), which uses the same architecture of BERT and is trained on Wikipedia corpora with 104 widely used languages. It is one of the most effective methods

---

[3] https://github.com/google-research-datasets/paws/tree/master/pawsx

| Dataset | Method | en | ar | bg | de | el | es | fr | hi | ru | sw | th | tr | ur | vi | zh | ja | ko | AVG |
|---|---|---|---|---|---|---|---|---|---|---|---|---|---|---|---|---|---|---|---|
| PAWS-X | mbert* | 94.0 | - | - | 85.7 | - | 87.4 | 87.0 | - | - | - | - | - | - | - | 77.0 | 73.0 | 69.6 | 82.0 |
| | mbert+wTD | **94.1** | - | - | 86.4 | - | 88.3 | 87.1 | - | - | - | - | - | - | - | 79.7 | 75.8 | 74.9 | 83.8 |
| | Syn.♡ | 94.0 | - | - | **87.8** | - | 85.9 | **89.1** | - | - | - | - | - | - | - | 80.7 | 75.8 | 76.3 | 84.3 |
| | X-MAML♡ | 94.0 | - | - | 86.5 | - | 87.6 | 87.3 | - | - | - | - | - | - | - | 80.3 | 76.2 | 76.5 | 84.1 |
| | mbert-adv | **94.1** | - | - | 86.5 | - | 89.2 | 88.0 | - | - | - | - | - | - | - | 79.9 | 75.4 | 76.5 | 84.2 |
| | Macedon-NoCM | 93.8 | - | - | 87.0 | - | **89.3** | 88.5 | - | - | - | - | - | - | - | **81.3** | 77.1 | 77.0 | 84.9 |
| | Macedon | 93.9 | - | - | 87.0 | - | 89.2 | 88.8 | - | - | - | - | - | - | - | **81.3** | **78.3** | **77.8** | **85.2** |
| QADSM | mbert* | 68.3 | - | - | 60.3 | - | - | 64.1 | - | - | - | - | - | - | - | - | - | - | 64.2 |
| | mbert+wTD | 68.0 | - | - | 60.9 | - | - | 62.8 | - | - | - | - | - | - | - | - | - | - | 63.9 |
| | Syn.♡ | 68.4 | - | - | 60.8 | - | - | 64.0 | - | - | - | - | - | - | - | - | - | - | 64.4 |
| | X-MAML♡ | 68.2 | - | - | 63.7 | - | - | 64.4 | - | - | - | - | - | - | - | - | - | - | 65.4 |
| | mbert-adv | 67.7 | - | - | 59.1 | - | - | 64.6 | - | - | - | - | - | - | - | - | - | - | 63.8 |
| | Macedon-NoCM | 67.3 | - | - | 63.8 | - | - | 63.0 | - | - | - | - | - | - | - | - | - | - | 64.7 |
| | Macedon | **68.6** | - | - | **64.0** | - | - | **64.9** | - | - | - | - | - | - | - | - | - | - | **65.8** |
| XNLI | mbert* | 80.8 | 64.3 | 68.0 | 70.0 | 65.3 | 73.5 | 73.4 | 58.9 | 67.8 | 49.7 | 54.1 | 60.9 | 57.2 | 69.3 | 67.8 | - | - | 65.4 |
| | mbert+wTD | 81.0 | 66.5 | 70.4 | 72.4 | 68.0 | 74.6 | 74.2 | 64.1 | 70.0 | 51.3 | 58.3 | 63.3 | 61.4 | 71.4 | 71.7 | - | - | 67.9 |
| | Syn.♡ | 81.6 | 65.4 | 69.3 | 70.7 | 66.5 | 74.1 | 73.2 | 60.5 | 68.8 | - | - | 62.4 | 58.7 | 69.9 | 69.3 | - | - | - |
| | X-MAML♡ | 82.7 | 68.4 | 72.8 | 74.1 | 70.7 | 76.5 | 76.0 | 65.8 | 72.1 | 59.9 | 62.5 | 65.7 | 64.6 | 73.9 | 74.9 | - | - | 70.7 |
| | mbert-adv | 82.6 | 66.8 | 71.1 | 72.3 | 69.1 | 75.2 | 74.2 | 64.2 | 70.2 | 51.7 | 57.2 | 63.1 | 60.9 | 72.6 | 71.6 | - | - | 68.2 |
| | Macedon-NoCM | 81.1 | 66.2 | 70.6 | 72.7 | 68.5 | 74.7 | 73.9 | 63.1 | 70.3 | 51.9 | 56.8 | 64.0 | 60.6 | 71.5 | 71.6 | - | - | 67.8 |
| | Macedon | 82.1 | 66.6 | 70.6 | 73.3 | 68.6 | 75.1 | 74.6 | 64.3 | 70.8 | 52.2 | 58.1 | 63.6 | 60.9 | 72.7 | 71.6 | - | - | 68.3 |
| | Macedon 2.5k | **82.9** | **69.2** | **74.6** | **75.0** | **70.7** | **76.7** | **77.3** | **67.2** | **73.4** | 59.9 | **63.5** | **67.8** | **65.1** | **74.5** | **75.7** | - | - | **71.6** |

Table 1: Performance comparison on the three datasets. "AVG" means the average accuracy of all languages. The highest scores per category are in **bold**. Results of ⋆ are taken from (Liang et al., 2020), and results of * are taken from (Hu et al., 2020). Results of ♡ are taken from (Nooralahzadeh et al., 2020) and (Ahmad et al., 2021) or obtain based on their official code release.

| Dataset | # of languages | Task | $|Train|^{all}$ | $|Dev|^{all}$ | $|Test|^{avg}$ |
|---|---|---|---|---|---|
| PAWS-X | 7 | PI | 50,004 | 14,000 | 14,000 |
| QADSM | 3 | QADSM | 100,200 | 30,000 | 10,000 |
| XNLI | 15 | NLI | 425,202 | 2,490 | 5,010 |

Table 2: Statistics of datasets.

for cross-lingual natural language understanding tasks.

- mbert-wTD (Hu et al., 2020) fine-tunes mbert with both English data and few-shot data, which can be considered as a multi-task training and is a strong baseline.

- Syn. (Ahmad et al., 2021), one of the state-of-the-art approaches for cross-lingual NLU tasks, takes advantage of the universal dependency tree structure in mbert to benefit cross-lingual tasks. We use the official release of Syn. implementation [4].

- X-MAML (Nooralahzadeh et al., 2020) proposes a cross-lingual meta learning architecture based on MAML (Finn et al., 2017) to learn a good initialization which can be adapted for other languages easily. It has shown its effectiveness on PI and NLU tasks. We adopt their official release of X-MAML implementation [5].

- mbert+adv (Chen et al., 2018b) learns language invariant representation based on adversarial train-

[4] https://github.com/wasiahmad/Syntax-MBERT
[5] https://github.com/copenlu/X-MAML

ing with generative adversarial networks (Goodfellow et al., 2014).

- Macedon-NoCM, a variant of Macedon, does not use covariance matrix modification.

### 5.1.3 Evaluation and Implementation Details

We use Accuracy to evaluate paraphrase identification, query advertisement matching, and natural language inference, following previous work (Hu et al., 2020; Liang et al., 2020). In implementation details are shown in the appendix.

### 5.2 Performance Comparison

In this section, we report the results of baselines and the proposed Macedon in Table 1.

First, according to Table 1, we find that the proposed Macedon outperforms all the state-of-the-art methods on three datasets with respect to all language average accuracy. Compared to Syn., it is a data augmentation method. It augments source language data to simulate target domain data as closely as possible. However, if the quality of the data is not high enough, it is difficult to align high-resource language and low-resource languages very well. However, the proposed Macedon does not rely on the augmented data. Compared to X-MAML, which is different from Syn., it is one of the state-of-the-art methods based on meta learning. However, based on Nooralahzadeh et al.

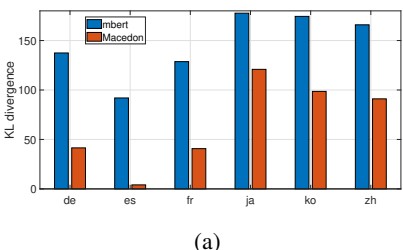 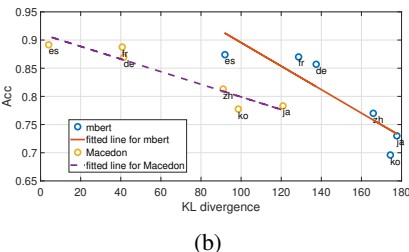

(a)              (b)

Figure 2: Figure 2(a): KL-divergence between different low-resource languages and English. Figure 2(b): Accuracy of different languages with respect to the KL-divergence between them and English.

(2020), X-MAML still needs hundreds or thousands low-resource language data to align them. When there are extremely limited low-resource language data, the proposed Macedon can be more effective to align different language data.

Second, fine-tuning mbert with both English data and limited target language data may only provide very little benefit for accuracy. Compared to mbert, mbert+wTD only achieves improvement on PAWS-X and XNLI dataset. On QADSM dataset, mbert+wTD even performs worse than that of mbert.

Third, covariance modification is effective to improve the performance of Macedon. Compared to Macedon-NoCM, Macedon has better performance on the three datasets. Because low-resource language data is limited, which is much smaller than the representation embedding, the estimated empirical covariance matrix is ill-conditioned. However, the covariance modification makes the empirical covariance matrix positive definite and provides a more accurate estimation under limited data. Therefore, the proposed Macedon shows better performance than Macedon-NoCM.

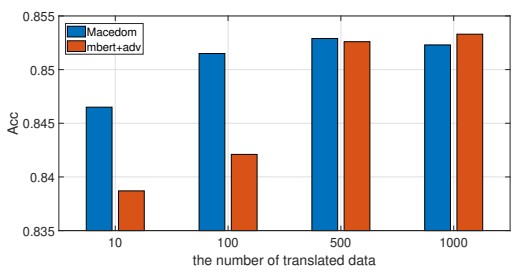

Figure 3: Accuracy of Macedon and adversarial training with different number of translated data

### 5.3 Macedon and Adversarial Training

In this section, we study the relationship between the proposed Macedon and mbert+adv (adversarial training method) to answer the second research question. We show the accuracy of Macedon and mbert+adv with different number of few-shot data in Figure 3.

According to Figure 3, we find that both Macedon and mbert+adv can have better performance as the number of few-shot data increases. With the increment of few-shot data, both Macedon and mbert-adv can achieve richer useful supervision to align different languages, so they can have better performance. On the other hand, when the translated data is limited, such as 10, 100, Macedon significantly outperforms than mbert+adv. The possible reason is that the discriminator of mbert+adv suffers from overfitting and can not distinguish different languages, so it can not provide effective supervision signal to update the classifier. However, when the amount of few-shot data increases to 1000, mbert+adv has better performance than that of Macedon. When mbert-adv can achieve enough target domain data, it can align languages better than Macedon, because Macedon just aligns the second-order moment (covariance matrix). In practical use, if there is only limited low-resource data, we suggest using Macedon.

### 5.4 Effectiveness of Aligning Different Languages

In this section, we show the empirical KL-divergence (Pérez-Cruz, 2008) between target languages and English representation on PAWS-X dataset in Figure 2 to answer the third research question.

Based on Figure 2(a), we find that the proposed Macedon is much more effective to align low-resource languages and high-resource languages than mbert. The KL-divergence between all low-resource languages and English of Macedon is significantly smaller than that of mbert. It shows that the distributions of low-resource languages and English representation is very close, which can

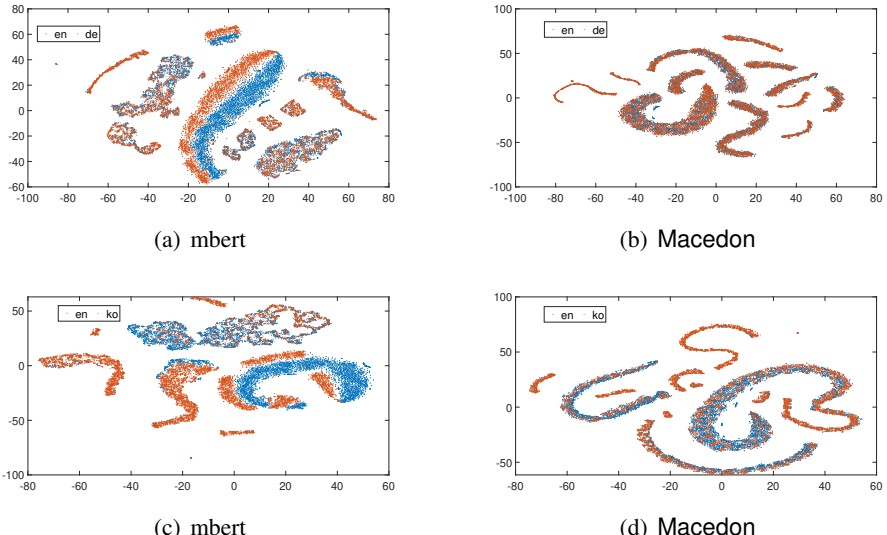

(a) mbert                 (b) Macedon

(c) mbert                 (d) Macedon

Figure 4: Representation visualization via t-SNE on PAWS-X . Figure 4(a) is the visualization of mbert on English and German; Figure 4(b) is the visualization of Macedon on English and German; Figure 4(c) is the visualization of mbert on English and Korean; Figure 4(d) is the visualization of Macedon on English and Korean.

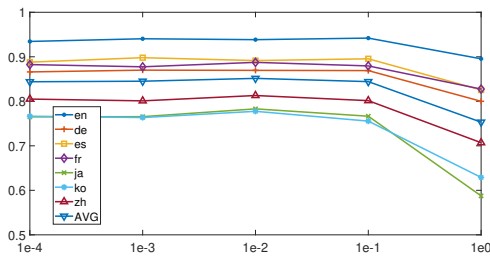

Figure 5: Accuracy with different values of coefficients.

provide explanation that Macedon shows higher accuracy than mbert.

The accuracy of different low-resource languages is almost inversely proportional to the KL-divergence according to Figure 2(b). For both the Macedon and mbert, if the KL-divergence between one low-resource language and English is small, the low-resource language may have high accuracy, vice versa. Because English has rich data, model usually can predict accurately on English dataset. Therefore, if the low-resource language representation is close to English representation, it can borrow more information from English to make accurate predictions. We also find one more interesting phenomenon. Usually, if a low-resource language is closer to English with respect to linguistic distance, it may be aligned to English better (Nooralahzadeh et al., 2020; Zhao et al., 2020). For example, Spanish, French, and German have shorter linguistic distance to English than Chinese, Korean, and Japanese, and Spanish, French, and German have smaller KL-divergence. One poten-

tial reason is that if the language has shorter linguistic distance to English, it will share more characteristics with English and be easier to align.

### 5.5 Sensitivity w.r.t. Hyper-parameter

In this section, we show the accuracy of Macedon with different values of coefficients on PAWS-X dataset in Figure 5 to answer the fourth research question. According to the Figure 5, we can find that for the averaged accuracy, it has similar values when changing the coefficient from $1e-4$ to $1e-1$. However, the accuracy will drop a lot when the coefficient is $1e0$. When the coefficient becomes $1e0$, the loss function focuses too much on the maximizing coding rate-distortion while ignoring the classification loss. Therefore, it will lead to the decline in model accuracy.

### 5.6 Case Study

In this section, we visualize the representation of Macedon and mbert respectively via t-SNE (Van der Maaten and Hinton, 2008) on PAWS-X dataset in Figure 4 to show how different language representation distributes. In Figure 4, there are two language pairs: 1) English and German in Figure 4(b), and 2) English and Korean in Figure 4(d). Figure 4 shows that the representation of mbert on English and German is not aligned well. Most of data with different languages can be classified easily. And this phenomenon is also shown on English and Korean data for mbert. However, for the proposed Macedon, its representation is mixed well among different languages, which intuitively

shows it can align different languages effectively.

# 6 Conclusion

In this paper, we study a practical scenario, i.e., learning a model with abundant high-resource language data and limited low-resource language data, for cross-lingual NLU tasks. We follow the general idea of aligning different languages, but different from widely used adversarial training models, we propose a novel approach named Macedon, which minimizes representation coding rate reduction among different languages. The motivation is to make model not to use extra codes to encode language information. The advantage compared to adversarial training methods is that the proposed Macedon does not rely on a large amount of low-resource language data because it does not need to estimate data distribution accurately. We conduct extensive experiments on three public datasets including paraphrase identification, natural language inference, and query advertisement matching. Experiment results show that the proposed Macedon outperforms the state-of-the-art baselines.

## Limitations

The proposed Macedon may be computational intensive in some cases. The proposed method needs to compute two terms $\hat{R}(\boldsymbol{X}, \epsilon)$ and $\hat{R}^c(\boldsymbol{X}, \epsilon|\Pi)$. In the two terms, the complexity of determinant function might be high. If the matrix is a $d$-dimensional square matrix, the complexity is $\mathcal{O}(d^\omega)$, where $\omega \geq 2$. Therefore, when representation dimensionality is high, such as XLM-RoBERTa-XL (Goyal et al., 2021), or the number of languages is very large (over hundreds or thousands of languages), the computation complexity of the proposed method may be high. How to reduce its computation complexity is our future work in the next step.

## Ethics Statement

Although our framework makes significant advancements in the field of cross-lingual natural language understanding, it also raises concerns about the societal impacts of automation, including job loss for those who work in annotation services and other industries that rely on human labor.

## Acknowledgement

This work is supported in part by the US National Science Foundation under grant NSF IIS-2226108. Any opinions, findings, and conclusions or recommendations expressed in this material are those of the author(s) and do not necessarily reflect the views of the National Science Foundation.

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

## A  Proof of Proposition 1

*Proof.* For any non-zero vector $\boldsymbol{v}$, we have

$$(1-\alpha)\boldsymbol{v}^T\boldsymbol{X}\boldsymbol{X}^T\boldsymbol{v} + \alpha\frac{tr(\boldsymbol{X}\boldsymbol{X}^T)}{d}\boldsymbol{v}^T\boldsymbol{v}$$
$$=(1-\alpha)(\boldsymbol{X}^T\boldsymbol{v})^T(\boldsymbol{X}^T\boldsymbol{v}) + \alpha\frac{tr(\boldsymbol{X}\boldsymbol{X}^T)}{d}\boldsymbol{v}^T\boldsymbol{v}.$$

Because $(\boldsymbol{X}^T\boldsymbol{v})^T(\boldsymbol{X}^T\boldsymbol{v}) \geq 0$, and $\frac{tr(\boldsymbol{X}\boldsymbol{X}^T)}{d} > 0$, we have $(1-\alpha)\boldsymbol{v}^T\boldsymbol{X}\boldsymbol{X}^T\boldsymbol{v} + \alpha\frac{tr(\boldsymbol{X}\boldsymbol{X}^T)}{d}\boldsymbol{v}^T\boldsymbol{v} > 0$. Therefore, $(1-\alpha)\boldsymbol{X}\boldsymbol{X}^T + \alpha\frac{tr(\boldsymbol{X}\boldsymbol{X}^T)}{d}\boldsymbol{I}$ is positive definite. $\square$

## B  Proof of Proposition 2

*Proof.* Because $\log\det(\cdot)$ is strictly concave, for $\beta_j > 0$, $\sum_{j=1}^k = 1$ and $\boldsymbol{Z}_j \in \mathbb{S}_{++}^d$, we have

$$\log\det(\sum_{j=1}^k \beta_j\boldsymbol{Z}_j) \geq \sum_{j=1}^k \beta_j\log\det(\boldsymbol{Z}_j), \quad (5)$$

and the equality holds iff $\boldsymbol{Z}_1 = ... = \boldsymbol{Z}_k$. Here, we take $\beta_j = \frac{n_j}{n}$ and $\boldsymbol{Z}_j = \boldsymbol{I} + \frac{d}{n_j\epsilon}((1-\alpha)\boldsymbol{X}_j\boldsymbol{X}_j^T + \alpha\frac{tr(\boldsymbol{X}_j\boldsymbol{X}_j^T)}{d}\boldsymbol{I})$. Therefore, we have

$$\sum_{j=1}^k \beta_j\boldsymbol{Z}_j$$
$$=\boldsymbol{I} + \frac{d}{n\epsilon}((1-\alpha)\sum_{j=1}^k \boldsymbol{X}_j\boldsymbol{X}_j^T$$
$$+\alpha\frac{tr(\sum_{j=1}^k \boldsymbol{X}_j\boldsymbol{X}_j^T)}{d}\boldsymbol{I}).$$

Because $\sum_{j=1}^k \boldsymbol{X}_j\boldsymbol{X}_j^T) = \boldsymbol{X}\boldsymbol{X}^T$, we have

$$\hat{R}(\boldsymbol{X},\epsilon) = \log\det(\sum_{j=1}^k \beta_j\boldsymbol{Z}_j), \quad (6)$$

$$\hat{R}^c(\boldsymbol{X},\epsilon|\Pi) = \sum_{j=1}^k \beta_j\log\det(\boldsymbol{Z}_j). \quad (7)$$

Therefore, the proposition holds. $\square$

## C  Implementation Details

For batch size and learning rate, we tune them on a validation set with grid search over $\{8, 16, 32\}$ and $\{1e-5, 2e-5, 3e-5\}$ respectively. For $\eta$, we tune it on a validation set with grid search over $\{1e-4, ..., 1e0\}$. For all experiments, we run 3 times and report the average results. We run our experiments on 4 NVIDIA RTX A6000.