# OpenReview forum: "Macedon: Minimizing Representation Coding Rate Reduction for Cross-Lingual Natural Language Understanding"
_EMNLP/2023/Conference — EMNLP 2023 Findings_

### Official Review · Reviewer_C8eb · 2023-08-05

**Soundness:** 3

**Excitement:**

3: Ambivalent: It has merits (e.g., it reports state-of-the-art results, the idea is nice), but there are key weaknesses (e.g., it describes incremental work), and it can significantly benefit from another round of revision. However, I won't object to accepting it if my co-reviewers champion it.

**Paper Topic And Main Contributions:**

This paper proposes an approach called Macedon for cross-lingual natural language understanding (NLU) tasks that aims to overcome the challenges of training models with limited low-resource language data. The paper addresses the problem of learning language-agnostic representations that can be used for cross-lingual transfer learning, which is important for many NLU tasks.

The main contributions of the paper are:

1. The introduction of Macedon, a new approach that uses a coding rate reduction technique to minimize the loss of information during representation learning, and adversarial training to align different language representations.

2. The evaluation of Macedon on three public cross-lingual benchmark datasets, including PAWS-X, QADSM, and XNLI, which shows that Macedon outperforms existing state-of-the-art methods for cross-lingual NLU.



**Questions For The Authors:**


1.What is the difference between mbert and Macedon-NoCM in the Table-1?
2. See the weaknesses.

**Reasons To Accept:**

The strengths of this paper includes

1. Novel approach: The paper proposes a novel approach called Macedon that uses a coding rate reduction technique to minimize the loss of information during representation learning to align different language representations.

2. Evaluation: The paper evaluates Macedon on three public cross-lingual benchmark datasets and compares it with existing state-of-the-art methods.

3. Contributions: The paper makes several contributions, including the proposal of two criteria for representation learning, the introduction of Macedon, and the evaluation of Macedon on benchmark datasets.


**Reasons To Reject:**

The potential weaknesses of this paper includes:

1.No mention about the base model architecture of the Macedon in this paper. Does it follow mbert? If yes, then why mbert was chosen? XLM-Roberta perfroms better than mbert in most of the tasks. Why it was not selected?

2.Selection of baselines are also not clear.

**Reproducibility:**

3: Could reproduce the results with some difficulty. The settings of parameters are underspecified or subjectively determined; the training/evaluation data are not widely available.

**Reviewer Confidence:**

4: Quite sure. I tried to check the important points carefully. It's unlikely, though conceivable, that I missed something that should affect my ratings.

---

> ### Author Rebuttal · Authors · 2023-08-29
>
> >No mention about the base model architecture of the Macedon in this paper. Does it follow mbert? If yes, then why mbert was chosen? XLM-Roberta perfroms better than mbert in most of the tasks. Why it was not selected?
>
> Yes, we selected mBERT as the backbone for our paper. Given that the proposed Macedon is a versatile framework applicable to various backbones, we specifically chose one of the most representative multilingual pretrained language models, mBERT, to demonstrate the effectiveness of our framework. Additionally, we introduced XLM-R-base as another backbone and presented the results for PAWS-X and XMLI in Tables A and B, respectively. Notably, our findings indicate that the proposed Macedon consistently outperforms the baselines when utilizing the XLM-R backbone.
>
> Table A: Performance comparison on the PAWS-X dataset with XLM-R-base as the backbone. The highest scores per category are in bold.
> |          |en   | de   | es   | fr   | ja   | ko   | zh   | AVG  |
> |-----------|------|------|------|------|------|------|------|------|
> |XLM-R     | 94.6 | 87.5 | 89.0 | 89.4 | 75.5 | 74.6 | 80.4 | 84.4 |
> |XLM-R+wTD | 94.1 |87.1 | 89.3 | 88.7 | 77.4 | 76.5 | 80.4 | 84.8|
> |XLM-R-adv | 94.5 | 87.1 | **89.5**| 89.1 | 77.2 | 77.3 | 80.7 | 85.1 |
> |Macedon   | **94.8** | **88.7** | 89.3 | **90.1** | **77.9** | **78.0** | **81.3** | **85.7** |
>
> Table B: Performance comparison on the XNMI dataset with XLM-R-base as the backbone. The highest scores per category are in bold.
> |           | en   | ar   | bg   | de   | el   | es   | fr   | hi   |
> |-----------|------|------|------|------|------|------|------|------|
> | XLM-R     | 84.8 | 71.7 | 77.9 | 77.5 | 76.2 | 79.1 | 78.2 | 70.1 |
> | XLM-R+wTD | 85.0 | 72.1 | 78.4 | 77.2 | 76.4 | 79.2 | 79.1 | 71.7 |
> | XLM-R-adv | 84.8 | 73.2 | 78.5 | 77.6 | 76.5 | 78.6 | 78.5 | 71.8 |
> | Macedon   | **85.3** | **73.5** | **79.0** | **78.8** | **77.6** | **80.9** | **79.6** | **72.0** |
> |           | **ru**  | **sw**   | **th**   | **tr**  | **ur**   | **vi**   | **zh**   |**AVG**  |
> | XLM-R     | 75.8 | 64.9 | 72.5 | 73.1 | 66.5 | 75.0 | 73.4 | 74.4 |
> | XLM-R+wTD | 76.3 | 66.5 | 73.9 | **74.1** | 68.0 | 75.4 | 75.2 | 75.2 |
> | XLM-R-adv | 76.6 | 65.6 | 73.3 | 73.8 | 68.3 | 75.7 | 74.8 | 75.2 |
> | Macedon   | **77.0** | **66.6** | **74.5** | 73.9 | **68.8** | **76.6** | **76.1** | **76.0** |
>
> >Selection of baselines are also not clear.
>
> Thanks for pointing it out. We use following baselines: (1) zero-shot on mBERT, (2) mbert-wTD (fine-tuning mBERT with English and few-shot data), (3) Syn. (one of the state-of-the-art approaches for cross-lingual NLU tasks via data augmentation), X-MAML (one of the state-of-the-art approaches for cross-lingual NLU tasks based on meta-learning), and mbert-adv (a popular and strong baseline which learns language-invariant representation based on adversarial training with generative adversarial networks).
>
> >What is the difference between mbert and Macedon-NoCM in the Table-1
>
> Macedon-NoCM is a variant of the proposed Macedon. Macedon-NoCM utilizes Multilingual BERT as the backbone without incorporating covariance matrix modification. The comparison between Macedon and Macedon-NoCM constitutes an ablation study. The entry labeled "mbert" in the performance comparison table refers to zero-shot training on Multilingual BERT, achieved through fine-tuning Multilingual BERT with English data.

---

### Official Review · Reviewer_hT3i · 2023-08-10

**Soundness:** 3

**Excitement:**

3: Ambivalent: It has merits (e.g., it reports state-of-the-art results, the idea is nice), but there are key weaknesses (e.g., it describes incremental work), and it can significantly benefit from another round of revision. However, I won't object to accepting it if my co-reviewers champion it.

**Paper Topic And Main Contributions:**

This paper addresses the challenge of training cross-lingual natural language understanding (NLU) models in situations where there is a scarcity of data for low-resource languages.

The primary focus of the paper is to propose a novel approach called "Macedon" that aims to improve cross-lingual NLU performance in scenarios where low-resource languages are involved. The authors highlight that existing methods, which often rely on adversarial training and mutual information estimation to learn language-agnostic representations, can struggle when data is extremely limited for a particular language. Accurately estimating data distribution becomes challenging in such cases.

The key contribution of the paper is the introduction of the "Macedon" approach, which stands for "minimizing representation coding rate reduction." The proposed method seeks to eliminate language-associated information in a model's representations. This is done by minimizing the need for extra codes to encode language-related information, measured using the rate-distortion function. In other words, the goal is to create a model that can effectively represent language-neutral features without introducing additional complexity related to language-specific aspects.



**Reasons To Accept:**

The paper presents a novel approach to improving cross-lingual natural language understanding (NLU) performance, particularly in scenarios where data for low-resource languages is scarce.

The introduction of the Macedon approach is innovative. It proposes a novel method for learning cross-lingual representations that minimizes the reliance on extra codes to encode language-related information. This approach is unique in its focus on rate-distortion functions to guide the representation learning process.



**Reasons To Reject:**

 While the paper claims that the proposed Macedon approach outperforms state-of-the-art methods, the comparison might be limited to a specific set of existing approaches. A more comprehensive comparison across a wider range of methods, such as XLMR, would provide a more complete understanding of the proposed approach's performance.

**Reproducibility:**

3: Could reproduce the results with some difficulty. The settings of parameters are underspecified or subjectively determined; the training/evaluation data are not widely available.

**Reviewer Confidence:**

4: Quite sure. I tried to check the important points carefully. It's unlikely, though conceivable, that I missed something that should affect my ratings.

---

> ### Author Rebuttal · Authors · 2023-08-29
>
> >While the paper claims that the proposed Macedon approach outperforms state-of-the-art methods, the comparison might be limited to a specific set of existing approaches. A more comprehensive comparison across a wider range of methods, such as XLMR, would provide a more complete understanding of the proposed approach's performance.
>
> Thank you for pointing that out! Given that the proposed Macedon is a versatile framework applicable to various backbones, we opted to feature one of the most representative multilingual pre-trained language models, mBERT, in our paper to illustrate the effectiveness of our framework. Furthermore, we introduced XLM-R-base as an additional backbone and presented the results for PAWS-X and XMLI in Tables A and B, respectively. Notably, our findings demonstrate that the proposed Macedon consistently outperforms the baselines when utilizing the XLM-R backbone.
>
> Table A: Performance comparison on the PAWS-X dataset with XLM-R-base as the backbone. The highest scores per category are in bold.
> |          |en   | de   | es   | fr   | ja   | ko   | zh   | AVG  |
> |-----------|------|------|------|------|------|------|------|------|
> |XLM-R     | 94.6 | 87.5 | 89.0 | 89.4 | 75.5 | 74.6 | 80.4 | 84.4 |
> |XLM-R+wTD | 94.1 |87.1 | 89.3 | 88.7 | 77.4 | 76.5 | 80.4 | 84.8|
> |XLM-R-adv | 94.5 | 87.1 | **89.5**| 89.1 | 77.2 | 77.3 | 80.7 | 85.1 |
> |Macedon   | **94.8** | **88.7** | 89.3 | **90.1** | **77.9** | **78.0** | **81.3** | **85.7** |
>
> Table B: Performance comparison on the XNMI dataset with XLM-R-base as the backbone. The highest scores per category are in bold.
> |           | en   | ar   | bg   | de   | el   | es   | fr   | hi   |
> |-----------|------|------|------|------|------|------|------|------|
> | XLM-R     | 84.8 | 71.7 | 77.9 | 77.5 | 76.2 | 79.1 | 78.2 | 70.1 |
> | XLM-R+wTD | 85.0 | 72.1 | 78.4 | 77.2 | 76.4 | 79.2 | 79.1 | 71.7 |
> | XLM-R-adv | 84.8 | 73.2 | 78.5 | 77.6 | 76.5 | 78.6 | 78.5 | 71.8 |
> | Macedon   | **85.3** | **73.5** | **79.0** | **78.8** | **77.6** | **80.9** | **79.6** | **72.0** |
> |           | **ru**  | **sw**   | **th**   | **tr**  | **ur**   | **vi**   | **zh**   |**AVG**  |
> | XLM-R     | 75.8 | 64.9 | 72.5 | 73.1 | 66.5 | 75.0 | 73.4 | 74.4 |
> | XLM-R+wTD | 76.3 | 66.5 | 73.9 | **74.1** | 68.0 | 75.4 | 75.2 | 75.2 |
> | XLM-R-adv | 76.6 | 65.6 | 73.3 | 73.8 | 68.3 | 75.7 | 74.8 | 75.2 |
> | Macedon   | **77.0** | **66.6** | **74.5** | 73.9 | **68.8** | **76.6** | **76.1** | **76.0** |

---

### Official Review · Reviewer_uCfL · 2023-08-10

**Soundness:** 4

**Excitement:**

4: Strong: This paper deepens the understanding of some phenomenon or lowers the barriers to an existing research direction.

**Paper Topic And Main Contributions:**

Multilingual language models (LMs) is a useful tool to tackle NLP tasks in the multilingual setting, or in a situation where there are no reliable pre-trained LMs available in the language of the desired tasks. However, one of the well-known bottlenecks of such multilingual LMs is the gap between a high-resource and low-resource language in performance when they are fine-tuned. The paper proposes a method to reduce this bias in multilingual LMs to ensure the performance in a low-resource language. The method is based on the coding rate function and the compression of mixed language representation is studied with the method. The claim is supported by the superior accuracy in multilingual language understanding tasks to the benchmark.

**Questions For The Authors:**

* The figures in general can be improved by increasing the font size.
* Why not compare the proposed method to XLM-R in addition to mBERT? I understand the method won't scale along with the model size, but at least XLM-R should fit, and it is worth to be added in the baseline given its popularity and superior performance to mBERT.

**Reasons To Accept:**

The approach is simple yet intuitive. There are a few works in this line of research but most of them focus on the vocabulary in a post-hoc manner, while this work focuses on the representation in each language, where they improve the latent space representation in terms of the rate distortion. Also, the paper is well-written, in a sense that one can understand the proposed method by just reading the paper with little struggle.

**Reasons To Reject:**

I understand the scope remains in language understanding, but would be nice to see how it works in more general setting like XTREME multilingual benchmark, or at least common multilingual tasks such as NER or QA. Also the experiment relies on mBERT as the baseline, which is less common in practice compared to XLM-R. Nevertheless, I am leaning toward accepting the paper even with lack of these aspects.

**Reproducibility:**

4: Could mostly reproduce the results, but there may be some variation because of sample variance or minor variations in their interpretation of the protocol or method.

**Reviewer Confidence:**

3: Pretty sure, but there's a chance I missed something. Although I have a good feel for this area in general, I did not carefully check the paper's details, e.g., the math, experimental design, or novelty.

---

> ### Author Rebuttal · Authors · 2023-08-29
>
> >I understand the scope remains in language understanding, but would be nice to see how it works in more general setting like XTREME multilingual benchmark, or at least common multilingual tasks such as NER or QA. Also the experiment relies on mBERT as the baseline, which is less common in practice compared to XLM-R. Nevertheless, I am leaning toward accepting the paper even with lack of these aspects.
>
> We sincerely appreciate your constructive suggestions. While we concur that evaluating our model in a more general setting such as XTREME would be beneficial, resource constraints have led us to select three representative datasets from both XTREME and XGLUE. Our intention is to include evaluations on XTREME and XGLUE in the revised version of our paper.
> Additionally, we have introduced an experiment utilizing XLM-R-base as the backbone, the results of which are presented in Table A and Table B for PAWS-X and XMLI, respectively. As demonstrated by the data in these tables, the proposed Macedon consistently outperforms the baseline models.
>
> Table A: Performance comparison on the PAWS-X dataset with XLM-R-base as the backbone. The highest scores per category are in bold.
> |          |en   | de   | es   | fr   | ja   | ko   | zh   | AVG  |
> |-----------|------|------|------|------|------|------|------|------|
> |XLM-R     | 94.6 | 87.5 | 89.0 | 89.4 | 75.5 | 74.6 | 80.4 | 84.4 |
> |XLM-R+wTD | 94.1 |87.1 | 89.3 | 88.7 | 77.4 | 76.5 | 80.4 | 84.8|
> |XLM-R-adv | 94.5 | 87.1 | **89.5**| 89.1 | 77.2 | 77.3 | 80.7 | 85.1 |
> |Macedon   | **94.8** | **88.7** | 89.3 | **90.1** | **77.9** | **78.0** | **81.3** | **85.7** |
>
> Table B: Performance comparison on the XNMI dataset with XLM-R-base as the backbone. The highest scores per category are in bold.
> |           | en   | ar   | bg   | de   | el   | es   | fr   | hi   |
> |-----------|------|------|------|------|------|------|------|------|
> | XLM-R     | 84.8 | 71.7 | 77.9 | 77.5 | 76.2 | 79.1 | 78.2 | 70.1 |
> | XLM-R+wTD | 85.0 | 72.1 | 78.4 | 77.2 | 76.4 | 79.2 | 79.1 | 71.7 |
> | XLM-R-adv | 84.8 | 73.2 | 78.5 | 77.6 | 76.5 | 78.6 | 78.5 | 71.8 |
> | Macedon   | **85.3** | **73.5** | **79.0** | **78.8** | **77.6** | **80.9** | **79.6** | **72.0** |
> |           | **ru**  | **sw**   | **th**   | **tr**  | **ur**   | **vi**   | **zh**   |**AVG**  |
> | XLM-R     | 75.8 | 64.9 | 72.5 | 73.1 | 66.5 | 75.0 | 73.4 | 74.4 |
> | XLM-R+wTD | 76.3 | 66.5 | 73.9 | **74.1** | 68.0 | 75.4 | 75.2 | 75.2 |
> | XLM-R-adv | 76.6 | 65.6 | 73.3 | 73.8 | 68.3 | 75.7 | 74.8 | 75.2 |
> | Macedon   | **77.0** | **66.6** | **74.5** | 73.9 | **68.8** | **76.6** | **76.1** | **76.0** |
>
> >The figures in general can be improved by increasing the font size.
>
> Thanks for your suggestion! We will modify them following your suggestion.

---

### Official Review · Reviewer_AJrg · 2023-08-10

**Soundness:** 3

**Excitement:**

3: Ambivalent: It has merits (e.g., it reports state-of-the-art results, the idea is nice), but there are key weaknesses (e.g., it describes incremental work), and it can significantly benefit from another round of revision. However, I won't object to accepting it if my co-reviewers champion it.

**Paper Topic And Main Contributions:**

The paper proposes a new method called Macedon for cross-lingual natural language understanding. Besides the vanilla cross-entropy loss, the paper introduces a regularization term that minimizes representation coding rate reduction between languages. The idea is to learn representations that do not use extra codes to encode language-specific information. Experiments on paraphrase, inference, and query matching tasks show Macedon outperforms baselines like mBERT.

**Reasons To Accept:**

- Minimizing coding rate reduction between languages is an interesting idea for aligning representations across languages. It does not require parallel data or accurate distribution estimates.

**Reasons To Reject:**

- The proposed model is not novelty enough, it simply applies the coding rate-distortion function to the representation of the last transformer layer. The authors should try to find which layer of the transformer should be applied to the coding rate-distortion function that can make the method more interesting.
- Lack of experiment. Authors mainly compare their model with mBERT and its variants without comparing it with current SOTA, e.g., XLM-R.

**Reproducibility:**

1: Could not reproduce the results here no matter how hard they tried.

**Reviewer Confidence:**

3: Pretty sure, but there's a chance I missed something. Although I have a good feel for this area in general, I did not carefully check the paper's details, e.g., the math, experimental design, or novelty.

**Typos Grammar Style And Presentation Improvements:**

- There are some typos in the paper, e.g., "high-source language data" in line 17 should be "high-resource language data", "When" -> "when" in line 22.
- Section 5.3 (RQ2) causes misunderstanding. Should it be the comparison with adversarial training?

---

> ### Author Rebuttal · Authors · 2023-08-29
>
> >The proposed model is not novelty enough, it simply applies the coding rate-distortion function to the representation of the last transformer layer. The authors should try to find which layer of the transformer should be applied to the coding rate-distortion function that can make the method more interesting.
>
> We believe there may be some misunderstandings regarding our methodology. In our paper, we aim to meet two key criteria for representations:
>
>     1. The learned representation should encompass sufficient information for the NLU task.
>     2. The distribution of representations across different languages should exhibit similarity.
>
> To fulfill the second criterion, we have extended and adjusted the structure of the coding rate-distortion function. Specifically, we advocate utilizing the final layer representation. This choice is essential to ensure that the representation employed for prediction adheres to the specified criterion. Consequently, we abstained from applying the proposed rate-distortion coding function to intermediary layers.
>
> >Lack of experiment. Authors mainly compare their model with mBERT and its variants without comparing it with current SOTA, e.g., XLM-R.
>
> Thanks for pointing it out! The proposed Macedon is a versatile framework with applicability to various backbones. In our paper, we have chosen to demonstrate the effectiveness of our framework using one of the most representative multilingual pre-trained language models, mBERT. Additionally, we have incorporated XLM-R-base as another backbone and presented the results in Table A and Table B for PAWS-X and XMLI, respectively. Notably, our results reveal that Macedon consistently outperforms the baselines when paired with the XLM-R backbone.
>
> Table A: Performance comparison on the PAWS-X dataset with XLM-R-base as the backbone. The highest scores per category are in bold.
> |          |en   | de   | es   | fr   | ja   | ko   | zh   | AVG  |
> |-----------|------|------|------|------|------|------|------|------|
> |XLM-R     | 94.6 | 87.5 | 89.0 | 89.4 | 75.5 | 74.6 | 80.4 | 84.4 |
> |XLM-R+wTD | 94.1 |87.1 | 89.3 | 88.7 | 77.4 | 76.5 | 80.4 | 84.8|
> |XLM-R-adv | 94.5 | 87.1 | **89.5**| 89.1 | 77.2 | 77.3 | 80.7 | 85.1 |
> |Macedon   | **94.8** | **88.7** | 89.3 | **90.1** | **77.9** | **78.0** | **81.3** | **85.7** |
>
> Table B: Performance comparison on the XNMI dataset with XLM-R-base as the backbone. The highest scores per category are in bold.
> |           | en   | ar   | bg   | de   | el   | es   | fr   | hi   |
> |-----------|------|------|------|------|------|------|------|------|
> | XLM-R     | 84.8 | 71.7 | 77.9 | 77.5 | 76.2 | 79.1 | 78.2 | 70.1 |
> | XLM-R+wTD | 85.0 | 72.1 | 78.4 | 77.2 | 76.4 | 79.2 | 79.1 | 71.7 |
> | XLM-R-adv | 84.8 | 73.2 | 78.5 | 77.6 | 76.5 | 78.6 | 78.5 | 71.8 |
> | Macedon   | **85.3** | **73.5** | **79.0** | **78.8** | **77.6** | **80.9** | **79.6** | **72.0** |
> |           | **ru**  | **sw**   | **th**   | **tr**  | **ur**   | **vi**   | **zh**   |**AVG**  |
> | XLM-R     | 75.8 | 64.9 | 72.5 | 73.1 | 66.5 | 75.0 | 73.4 | 74.4 |
> | XLM-R+wTD | 76.3 | 66.5 | 73.9 | **74.1** | 68.0 | 75.4 | 75.2 | 75.2 |
> | XLM-R-adv | 76.6 | 65.6 | 73.3 | 73.8 | 68.3 | 75.7 | 74.8 | 75.2 |
> | Macedon   | **77.0** | **66.6** | **74.5** | 73.9 | **68.8** | **76.6** | **76.1** | **76.0** |
>
> >There are some typos in the paper, e.g., "high-source language data" in line 17 should be "high-resource language data", "When" -> "when" in line 22. Section 5.3 (RQ2) causes misunderstanding. Should it be the comparison with adversarial training?
>
> Thank you for your valuable suggestions! We will incorporate these changes in the revised version.

---

### Meta-Review · Area_Chair_uXLe · 2023-09-09

**Recommendation:** 3

**Metareview:**

The paper addresses cross-lingual natural language understanding and proposes to minimize representation coding rate reduction between languages in order to learn representations that do not use extra codes to encode language-specific information.

Pros / Strengths:
- The idea is novel and interesting as the approach does not require parallel data or accurate distribution estimates.
- The paper is well written.
- The approach is evaluated on three datasets against some baseline models based on mbert.

Cons / Weaknesses:
- The baselines are not convincing (important models such as XML-R are missing), weakening the experimental sections. (Note: the authors have extended their experiments during the response period, including XML-R for at least two of their datasets. Why not all three?)
- The number of tasks is small considering that there are many more popular multilingual benchmark datasets available.

Action items for improving the paper:
- The experimental section is a bit weak. To increase the value of the paper for other researchers and its influence on future research, the authors should add results with stronger multilingual models (such as XLM-R (as already partly provided during the rebuttal phase)) as well as results on (at least a few) more datasets.

---

### Decision · Program_Chairs · 2023-10-07

**Decision:**

Accept-Findings

**Comment:**

The paper addresses cross-lingual natural language understanding and proposes to minimize representation coding rate reduction between languages in order to learn representations that do not use extra codes to encode language-specific information.

Pros / Strengths:
- The idea is novel and interesting as the approach does not require parallel data or accurate distribution estimates.
- The paper is well written.
- The approach is evaluated on three datasets against some baseline models based on mbert.

Cons / Weaknesses:
- The baselines are not convincing (important models such as XML-R are missing), weakening the experimental sections. (Note: the authors have extended their experiments during the response period, including XML-R for at least two of their datasets. Why not all three?)
- The number of tasks is small considering that there are many more popular multilingual benchmark datasets available.

Action items for improving the paper:
- The experimental section is a bit weak. To increase the value of the paper for other researchers and its influence on future research, the authors should add results with stronger multilingual models (such as XLM-R (as already partly provided during the rebuttal phase)) as well as results on (at least a few) more datasets.